# An SSHC Interface Circuit for Energy Harvesting of Piezoelectric Flags

**Yuansheng Chen** [1] , **Lichen Tong** [1] , **Jianzhou Du** [1] , **Hongli Ji** [2],* **and Pengcheng Zhao** [3]

1   School of Mechanical Engineering, Yancheng Institute of Technology, Yancheng 224051, China; chenys@ycit.edu.cn (Y.C.); tlc1998211@126.com (L.T.); dujianzhou123@163.com (J.D.)
2   State Key Laboratory of Mechanics and Control of Mechanical Structures, Nanjing University of Aeronautics and Astronautics, Nanjing 210024, China
3   School of Energy and Power Engineering, Nanjing University of Science and Technology, Nanjing 210094, China; zhaopengcheng1031@163.com
*   Correspondence: jihongli@nuaa.edu.cn; Tel.: +86-25-84891123

**Abstract:** Piezoelectric flags have functions of both classic flags and energy harvesting, and are becoming a new research focus. Interface circuits that convert wind energy to electrical energy are the key component of piezoelectric flags. A new structure for piezoelectric flags was designed to generate vibration by wind induction. After theoretical analysis, only SEH (standard energy harvesting) and SSHC (synchronized switch-harvesting-on capacitors) interface circuits were found suitable for piezoelectric flags. Simulation in Multisim was performed to compare SEH and SSHC in different load resistance. Experiments were carried out using different wind speeds. The on-time and delay-time of each switch were controlled by the proposed control algorithm. Both simulation and experimental results indicate that the output voltage with SSHC is higher than the output voltage with SEH. When the resistance is 1700 kΩ and the wind speed is 24 m/s, the output power of SSHC can be increased by 45.63% compared with the SEH circuit.

**Keywords:** wind power; piezoelectric flag; energy harvesting; SSHC





## 1. Introduction

With development of the Internet of Things, wireless sensing nodes are becoming increasingly more of an essential bridge connecting the real world and Internet. The energy sources to power the wireless sensing nodes are one of the key technologies in wireless sensing nodes. Because of the requirement of periodic recharging or replacement, the classic method using batteries is neither practical nor economical [1]. To manage this problem, harvesting environmental vibration energy such as piezoelectric micro wind power is receiving increasingly more attraction.

Piezoelectric micro wind power devices convert wind energy into vibration energy and obtain electrical energy through piezoelectric effect. It can harvest the environmental wind energy to power microelectronic devices. Owing to the advantages of softness, flexibility, and easy-cutting, piezoelectric polymers such as PVDF (polyvinylidene fluoride) are designed as a flag that has functions of both classic flags and energy harvesting. Umair et al. developed an inverted flag and harvest energy via a flapping motion [2]. Liu et al. [3] designed and tested a wind-induced flag-swing piezoelectric energy harvester (PEH). The piezoelectric cantilever beam swings by a flexible band that is vibrated by wind-induction. Zhang [4] proposed a piezoelectric energy harvesting circuit that consists of a piezoelectric element, synchronized switch-harvesting-on capacitors (SSHC) circuit, two sampling resistors, control circuit, full-bridge rectifier, and load. This circuit can accurately detect the zero-crossing time and automatically adjust the pulse width of the control signal according to the state of the circuit. Angelov [5] presented a novel architecture for realizing the synchronized switch-harvesting-on capacitors (SSHC) technique used for enhanced energy

extraction from piezoelectric transducers. Though the SSHC can harvest more energy under the same conditions, its application in piezoelectric flags is less reported.

According to the principle of wind-induced vibration, a new structure of piezoelectric flag was designed to convert wind energy into vibration energy. Piezoelectric film converts vibration energy into electrical energy. Interface circuits for energy harvesting of the designed piezoelectric flag is analyzed by theory, simulation, and experiments. Simulation with Multisim software was performed to compare the SEH and SSHC interface circuits for the designed piezoelectric flag. The difficulty of the SSHC interface circuit is the control of the switches, and experiments of the SSHC interface circuit are less reported. When the current through the piezoelectric flag reaches the zero-crossing, the switches in the SSHC circuit are turned on and off in a specific order. These switches, controlled by pluses, are generated by an Arduino Leonardo board with the proposed control algorithm. Since delay of zero-crossing detection is unavoidable, the delay time of switching is also discussed. The experiment was carried out to validate the effectiveness of the proposed SSHC interface circuit in different wind speed and different load resistance.

## 2. Structure of the Piezoelectric Flag

The piezoelectric flag was designed with PVDF (polyvinylidene fluoride) piezoelectric film, which is a kind of soft piezoelectric material having the advantage of softness and easy-cutting, as shown in Figure 1. When force or mechanical deformation is applied on the PVDF piezoelectric film, an electric charge is generated on its surfaces as the piezoelectric effect. In the case of a periodic force such as vibration, alternating current is generated by the PVDF piezoelectric film and it can be harvested by an interface circuit to power electric devices.

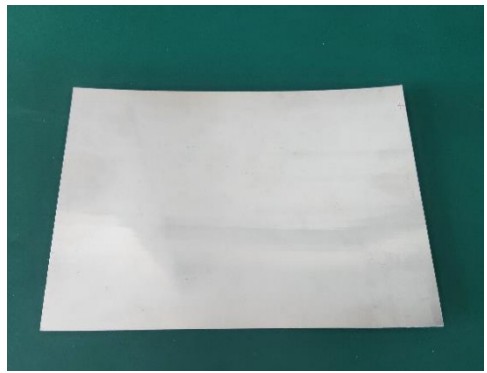

**Figure 1.** PVDF piezoelectric film.

The operational principle of the proposed piezoelectric flag is shown Figure 2. When the wind flows through a blunt body, a wind-induced vibration is generated on the blunt body of the piezoelectric flag [6]. Mechanical energy is then transferred into electric energy by piezoelectric elements. The interface circuit collects the electric energy and modulates the voltage to power the load or charge the battery [7]. Obviously, the structure and interface circuit are two important components of the proposed piezoelectric flag. The first generates more vibration, and the second gathers the electric energy [8].

The structure of the piezoelectric flag was designed with a blunt body and PVDF piezoelectric films, as shown in Figure 3a. The base between the two piezoelectric films was a steel plate 0.6 mm in thickness. The PVDF piezoelectric films were made of polyvinylidene fluoride whose size is 250 mm × 200 mm × 0.2 mm. As a key component for energy conversion, it can generate electric voltage between its two surfaces when vibrating. The two piezoelectric films were bonded on the both sides of the steel plate that was partly embedded in the blunt body. A steel cantilever beam of size 50 mm × 30 mm × 0.6 mm was designed to support the blunt body. One end of the cantilever beam was fixed on the experimental platform, and the other end was connected to the blunt body whose size is

250 mm × 30 mm × 30 mm. Because high-elastic manganese steel has the advantages of good elasticity and strong impact resistance, the cantilever beam was designed with high elastic manganese steel to vibrate the blunt body and piezoelectric film. When the wind flows from the opposite side of the piezoelectric films, a periodical force is generated on the blunt body, and the steel cantilever beam bends periodically; the base with the piezoelectric films then vibrates along the vertical direction of incoming flow, as shown in Figure 3b. The structure of the piezoelectric flag was inspired by classic cantilever beam and piezoelectric structure. Without the blunt body and steel cantilever, it is the same as a bimorph structure. The blunt body is an important part of piezoelectric flag, and it subjects periodic force in a suitable flow field. The periodic force generates the vibration, and the piezoelectric film converts the vibration energy into electric energy.

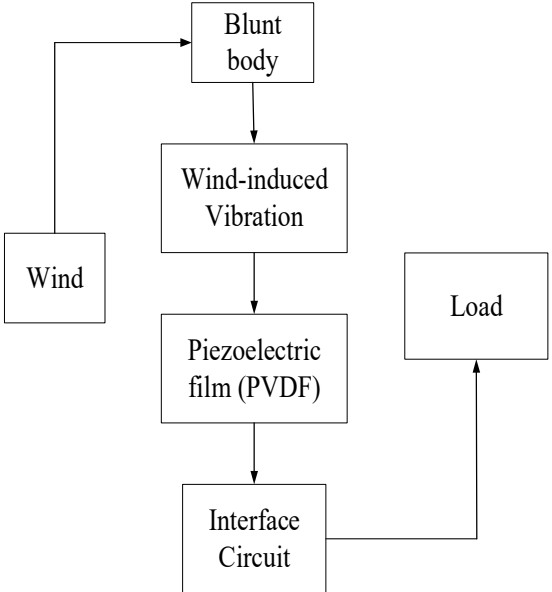

**Figure 2.** Principle of the proposed piezoelectric flag.

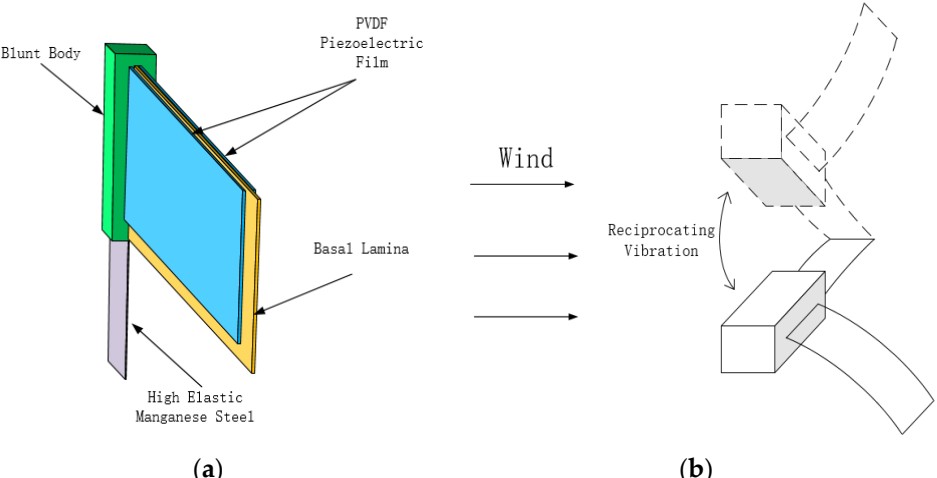

| (a) | (b) |

**Figure 3.** Structure and vibration direction of the piezoelectric flag: (**a**) structure of the proposed piezoelectric flag; (**b**) direction of wind and vibration.

## 3. SSHC Interface Circuit

### 3.1. Standard Energy Harvesting

The interface circuit is one of the key components to gather electric energy. There are several commonly used circuits in piezoelectric energy harvesting, such as SEH, SECE (synchronous electric charge extraction), SSHI (synchronized switch-harvesting-on inductor),

and SSHC [9–12]. SEH is a classic interface circuit for piezoelectric energy harvesting, and its diagram is shown in Figure 4. The electricity from piezoelectric elements is alternating current; a rectifier bridge consisting of four diodes convert alternating current to direct current. The capacitor $C$ stabilizes the voltage to power load $R$.

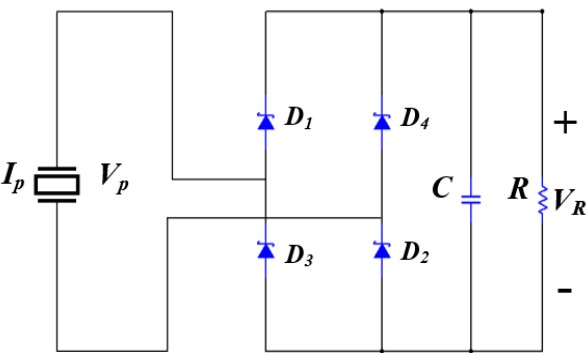

**Figure 4.** Diagram of the SEH interface circuit.

When the piezoelectric films are vibrated by wind, the charge is generated on their surfaces and the alternating current voltage $V_p$ changes with the vibration. Let $V_D$ represent the diode forward voltage drop; the bridge rectifier turns on when $|V_p| \geq 2V_D + V_R$. The bridge rectifier turns off and the power is not transferred to load $R$ in case of $|V_p| \leq 2V_D + V_R$. When the bridge rectifier is off, the energy from the piezoelectric films is lost because it is not transferred to capacitor $C$ or load $R$. To reduce the energy loss, an effective method is to accelerate the voltage $V_p$ flipping from $+V_p$ to $-V_p$. The faster the voltage $V_p$ flips, the shorter time that the bridge rectifier is off.

As air flows through the piezoelectric flag, the blunt body is vibrated by the alternating shedding of vortex. In this case, the displacement of the piezoelectric films satisfies the following equation:

$$u = U_M \sin(\omega t) \tag{1}$$

where $u$ is the displacement of piezoelectric flag, $U_M$ is the displacement maximum, and $\omega$ is angular frequency of the vibration. At time $t_o$, the displacement $u$ shown in Figure 5 achieves its maximum and the bridge rectifier turns off. The voltage of the piezoelectric flag begins to decrease from $|V_P|$. The displacement $u$ reaches its negative maximum and the bridge rectifier turns off and $|V_P| < (2V_D + V_R)$ at time $t_0 + T/2$. When $|V_P| > (2V_D + V_R)$, the bridge rectifier turns on. The piezoelectric flag starts to power the load $R$.

During the time interval $[t_0, t_0 + T/2]$, the variation in voltage on load $R$ is very small and can be ignored. Then, the charge flow through load $R$ [13] is:

$$-\int_{t_0}^{t_0+T/2} i\,dt = \frac{V_R}{R} \cdot \frac{T}{2} \tag{2}$$

The outflow current $I_P$ of the piezoelectric flag can be obtained:

$$I_p = \alpha \dot{u} - C_P \dot{V}_{C_P} \tag{3}$$

where $\alpha$ is representative of the force factor and $u$ is the vibration displacement of the piezoelectric flag. Substituting Equation (3) into Equation (2) gives:

$$V_R = \frac{2\alpha U_M \omega R}{2RC_0\omega + \pi} \tag{4}$$

where $C_0$ represents the capacitance value of the piezoelectric flag and $\omega$ represents the angular velocity of blunt body vibration.

The relationship between output power $P_{SEH}$ of the standard energy-harvesting interface circuit and load resistance $R$ can be obtained:

$$P_{SEH} = \frac{V_R^2}{R} = \frac{(2\alpha U_M \omega R)^2}{R(2RC_0\omega + \pi)} \tag{5}$$

Let $dP_{SEH}/dR = 0$; the optimal load corresponding to this circuit can be derived:

$$R_{opt} = \frac{\pi}{2C_0\omega} \tag{6}$$

Substituting Equation (6) into Equation (5), the maximum output power of the standard energy harvesting interface circuits is:

$$P_{SEH-M} = \frac{\omega \alpha^2 U_M^2}{2\pi C_0} \tag{7}$$

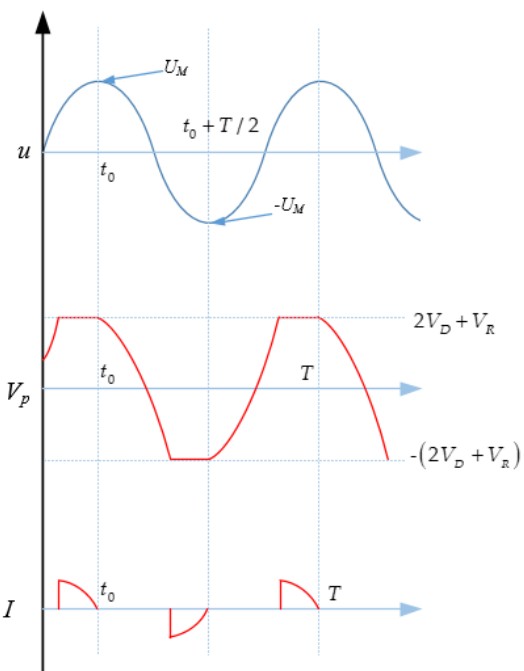

**Figure 5.** Theoretical waveform of the SEH circuit.

### 3.2. Principal of SSHC

Except for SEH and SSHC, an electrical inductor is an essential part of SECE and SSHI interface circuits. Because the piezoelectric elements can be approximately equivalent to a capacitor, it forms a resonance circuit with the connected inductor, and its period $T$ can be calculated by:

$$T = 2\pi\sqrt{LC_p} \tag{8}$$

where $L$ is the connected inductance and and $C_p$ is the capacitance of the piezoelectric element. According to the principle of SECE and SSHI [14], the turn-on time $t_{on}$ of switches should be half of the period $T$:

$$t_{on} = \pi\sqrt{LC_p} \tag{9}$$

It should be noted that the piezoelectric capacitance is relatively smaller, which is usually less than 20 nF. If the turn-on time $t_{on}$ is chosen to be 1 ms, the inductance $L$ is larger than 5.066$H$:

$$L \geq \frac{t_{on}^2}{\pi^2 C_p} = 5.066H \tag{10}$$

The power generated by piezoelectric films is limited, and the resistance of the connected inductor has to be very small to avoid power consumption. As a matter of fact, it is difficult to design an electrical inductor of 5.066$H$ with a smaller resistance. On the other hand, it is difficult to control the turn-on time to be less than 1 ms with a microcontroller of low power consumption.

An SSHC interface circuit without an inductor is proposed to harvest the energy of the piezoelectric flag. As shown in Figure 6, the proposed SSHC interface circuit consists of five switches, four diodes, and two capacitors. The right part with four diodes and a capacitor $C$ is the same as the SEH interface circuit, and the left part, five switches and capacitor $C_1$, was designed to raise the voltage generated by the piezoelectric films. These five switches are divided into three groups: (1) $S_{1A}$, $S_{1B}$; (2) $S_2$; and (3) $S_{3A}$, $S_{3B}$. Three pulse signals from the microcontroller are generated to drive these three groups of switches, which synchronously flip the voltage $V_1$ on capacitor $C_1$.

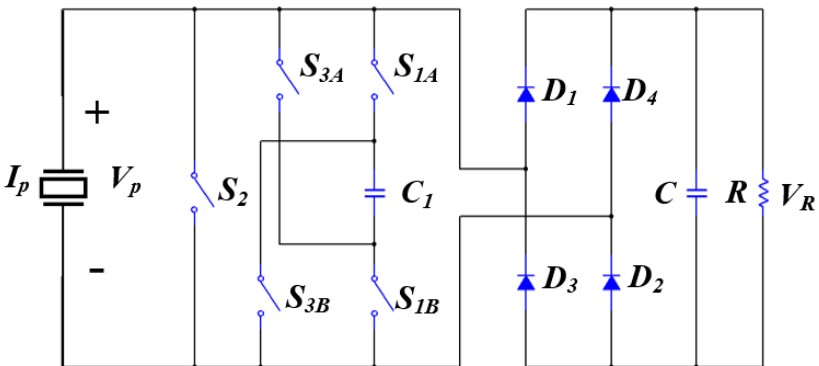

**Figure 6.** Diagram of the SSHC interface circuit.

When the current $I_p$ from the positive to negative side reaches the zero-crossing point, the switch $S_{1A}$, $S_{1B}$ turns on and some charge on the piezoelectric films flows to capacitor $C_1$. Next, switch $S_{1A}$, $S_{1B}$ turns off and switch $S_2$ turns on, which clears the remaining charge on the piezoelectric films. Then, switch $S_2$ turns off and switch $S_{3A}$, $S_{3B}$ turns on. In this case, some charge on $C_1$ flows back to the piezoelectric films in an opposite side. When the current $I_p$ from the negative to positive side reaches the zero-crossing point, similar switching is operated in the same order; a detailed procedure of SSHC can be found in [15]. Under the action of a switching capacitor, the voltage reversal is completed. Voltage flip action improves the recovery efficiency of the SSHC circuit.

The PVDF piezoelectric film is equivalent to a current source $I_p$, capacitor $C_p$, and resistor $R_p$, as shown in Figure 7. If the piezoelectric flag moves from zero to the maximum displacement, the voltage on the piezoelectric film is larger than zero, $V_p > 0$. When $V_P > V_{D_1} + V_{D_2}$, diode $D_1, D_2$ is in on-state and $D_3, D_4$ is in off-state, and current flows. At the same time, all switches are switched off. The SSHC circuit is equivalent to a full-bridge rectification circuit.

From the relationship between current and voltage in the circuit, the voltage of the piezoelectric film is:

$$V_P = V_{D_1} + V_{D_2} + V_R \tag{11}$$

$$I_P(t) + C_p \frac{dV_P}{dt} + \frac{V_P}{R_P} = C \frac{dV_R}{dt} + \frac{V_R}{R} \tag{12}$$

When the piezoelectric flag is about to move to the maximum forward displacement, the current $I_p$ from the negative to positive side reaches the zero-crossing point. Assuming voltage $V_{C_1} = 0$ on the switching capacitor, then the switch $S_{1A/B}$ is turned on. According to the law of conservation of charge, charges on the piezoelectric flags are split equally with the switching capacitor $C_1$ because $C_P = C_1$. At this time, the voltage of switching capacitor $C_1$ begins to increase from 0, and $V_{C_1} < V_{D_1} + V_{D_2} + V_R$. Current flow is shown in Figure 8.

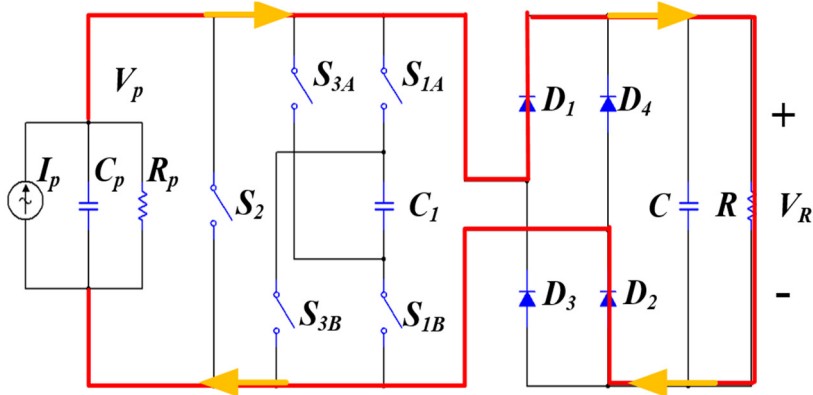

**Figure 7.** Current flow of the SSHC circuit when $V_p > 0$.

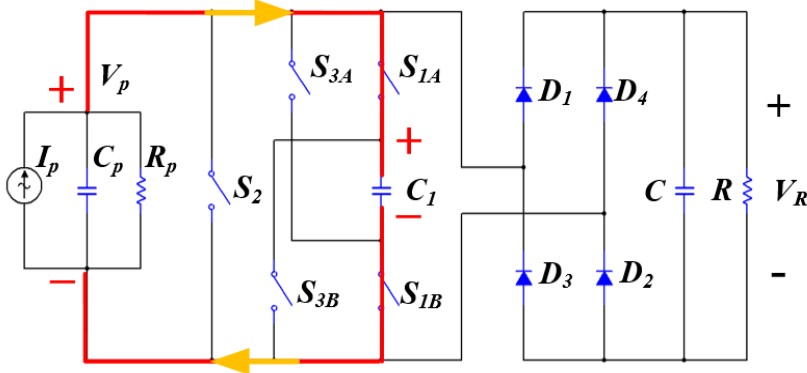

**Figure 8.** Current flow of the SSHC circuit when switch $S_{1A/B}$ is turned on ($V_{C_P} > V_{C_1}$).

The voltage of the piezoelectric film and switching capacitor is:

$$V_P = V_{C_1} = \frac{C_P}{C_P + C_1}\left(V_{D_1} + V_{D_2} + V_R\right) \tag{13}$$

Because of $C_P = C_1$, the equation above can be simplified:

$$V_P = V_{C_1} = \frac{1}{2}\left(V_{D_1} + V_{D_2} + V_R\right) \tag{14}$$

According to Kirchhoff's law, the current in the circuit is:

$$I_P(t) + C_P\frac{dV_P}{dt} + \frac{V_P}{R_P} = C_1\frac{dV_{C_1}}{dt} \tag{15}$$

Switch $S_{1A/B}$ is turned off soon after it was turned on. Switch $S_2$ is on at the same instant that switch $S_{1A/B}$ turns off, and the piezoelectric film is short-circuited. The voltage of the piezoelectric film is zero and the voltage on the switching capacitor remains. The current flow of this state is shown in Figure 9.

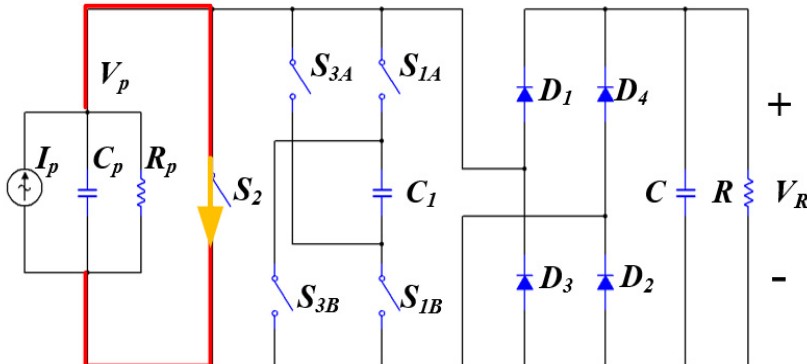

**Figure 9.** Current flow of the SSHC circuit when switch $S_2$ is turned on.

Switch $S_2$ is on for a very short time and switch $S_{3A/B}$ is turned on after $S_2$ is off. In this case, the capacitors $C_1$ and $C_p$ are connected in parallel, and the capacitor of piezoelectric film $C_p$ is charged because of $V_{C_1} > V_{C_P}$. Charges stored in switched capacitors $C_1$ are transferred into piezoelectric films, and the voltage on the PVDF piezoelectric film is less than zero, $V_P < 0$. The current flow of this state is shown in Figure 10.

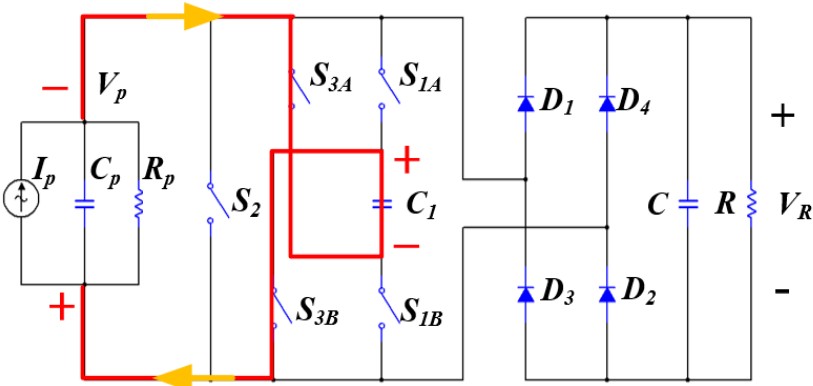

**Figure 10.** Current flow of the SSHC circuit when switch $S_{3A/B}$ is turned on ($V_{C_P} < V_{C_1}$).

In this case, the voltage on piezoelectric film is:

$$- V_P = V_{C_1} = \frac{C_1 C_P}{(C_P + C_1)^2} \left( V_{D_1} + V_{D_2} + V_R \right) \tag{16}$$

Its simplification is:

$$- V_P = V_{C_1} = \frac{1}{4} \left( V_{D_1} + V_{D_2} + V_R \right) \tag{17}$$

According to Kirchhoff's current law, the following equation can be obtained:

$$C_1 \frac{dV_{C_1}}{dt} = I_P(t) + C_P \frac{d(-V_P)}{dt} - \frac{V_P}{R_P} \tag{18}$$

Until then, the current $I_P$ reaches the zero-crossing point from the positive to negative side; voltage on the piezoelectric film changes from its maximum to zero, and then towards its negative voltage. The SSHC circuit completes the flip from maximum forward voltage to negative voltage. During this time range, the voltage on the piezoelectric film is less than zero, $V_P < 0$.

The piezoelectric film then begins to move to the negative maximum displacement; voltage and current begin to grow negatively and $V_P < 0$. When $-V_P > V_{D_3} + V_{D_4}$, diode $D_3$, $D_4$ is in on-state, and diode $D_1$, $D_2$ is in off-state. There is about a 0.7 V drop at both

ends of diode $D_3$, and the voltage of diode $D_2$ is $-(V_R + V_{D_1})$. All switches are in off-state and current is flowing, as shown in Figure 11.

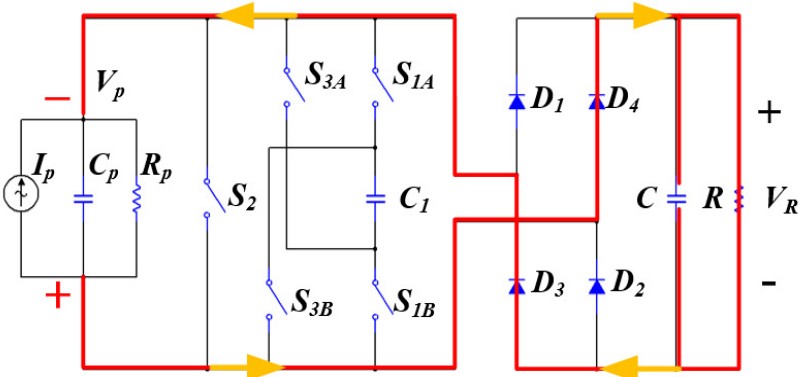

**Figure 11.** Current flow of the SSHC circuit when $V_P < 0$.

In this situation, the voltage on the piezoelectric film is:

$$-V_P = V_{D_3} + V_{D_4} + V_R \tag{19}$$

$$I_P(t) + C_p \frac{d(-V_P)}{dt} - \frac{V_P}{R_P} = C\frac{dV_R}{dt} + \frac{V_R}{R} \tag{20}$$

When the piezoelectric film moves to the maximum negative displacement, the voltage of the piezoelectric film is $V_P < 0$. The current $I_P$ is about to cross zero from negative to positive. The voltage of the switched capacitor is $V_{C_1} = (V_R + V_{D_1} + V_{D_2})/4$. Now, switch $S_{3A/B}$ is turned on, and capacitors $C_1$ and $C_p$ are connected in parallel. According to the law of conservation of charge, charges on the piezoelectric film are split equally with the switched capacitor $C_1$, because of $C_P = C_1$. The piezoelectric film charges the switched capacitor again until the voltage on the piezoelectric film is the same as that on the switched capacitor. The voltage of the switched capacitor $C_1$ begins to increase from $(V_R + V_{D_1} + V_{D_2})/4$ and $V_{C_1} < (V_R + V_{D_3} + V_{D_4})$. The current flow of this state is shown in Figure 12.

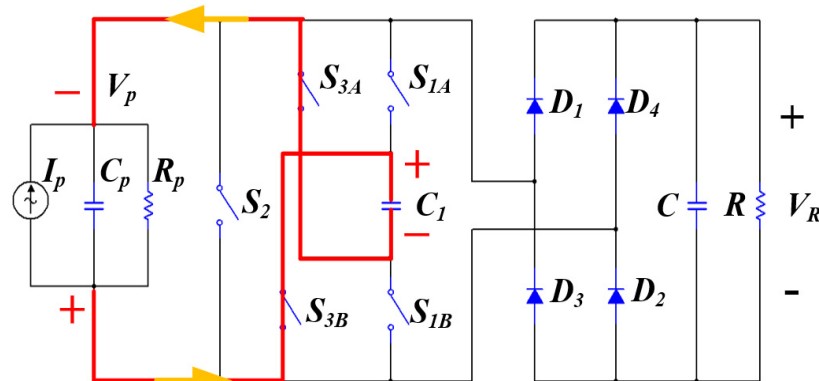

**Figure 12.** Current flow of the SSHC circuit when switch $S_{3A/B}$ is turned on ($V_{C_P} < V_{C_1}$).

In this case, the voltage of the piezoelectric film and switched capacitor is:

$$-V_P = V_{C_1} = \left(1 + \frac{1}{4}\right)\frac{C_1 C_P}{(C_P + C_1)^2}(V_{D_1} + V_{D_2} + V_R) \tag{21}$$

Simplification can be obtained:

$$-V_P = V_{C_1} = \frac{5}{8}(V_{D_1} + V_{D_2} + V_R) \tag{22}$$

The current of the SSHC circuit is:

$$I_P(t) + C_p \frac{d(-V_P)}{dt} - \frac{V_P}{R_P} = C_1 \frac{dV_{C_1}}{dt} \tag{23}$$

Then, switch $S_{3A/B}$ is turned off and switch $S_2$ is turned on; the piezoelectric film is short-circuited. The voltage of the piezoelectric flag is cleared and the voltage on the switching capacitor remains. The current flow is shown in Figure 13.

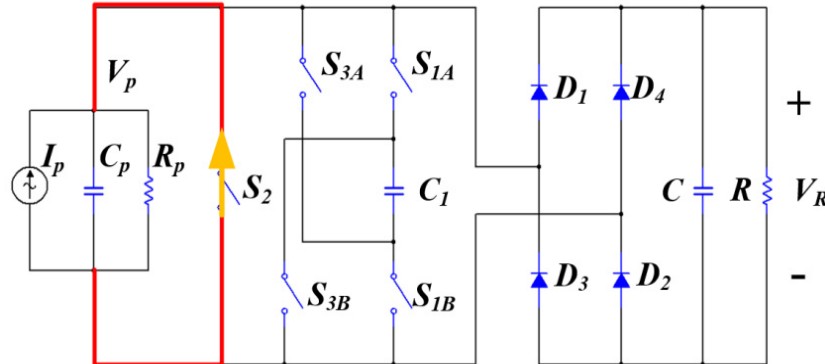

**Figure 13.** Current flow of the SSHC circuit when switch $S_2$ is turned on.

Switch $S_2$ is off soon after its on-state. Then, switch $S_{1A/B}$ is turned on after switch $S_2$ is off. Capacitors $C_1$ and $C_p$ are again connected in parallel. At this time, the voltage on the piezoelectric film is zero and capacitor $C_1$ applies a reverse voltage and charges the piezoelectric film. The charge stored on capacitor $C_1$ begins to transfer to the piezoelectric flag until $V_{C_P} = V_{C_1}$, which makes $V_P > 0$, as shown in Figure 14.

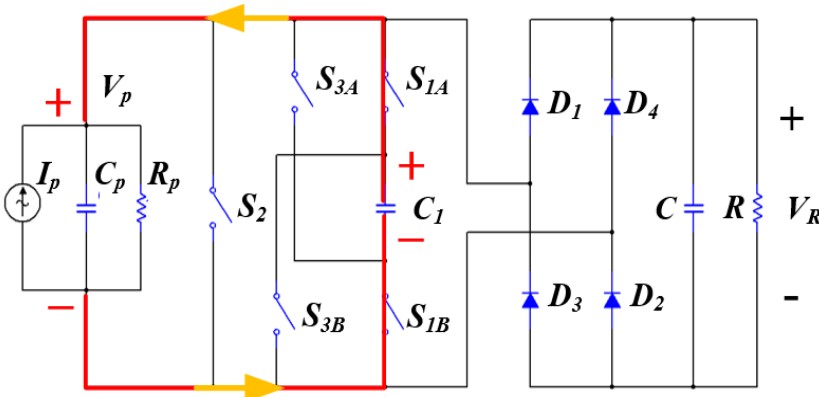

**Figure 14.** Current flow of the SSHC circuit when switch $S_{1A/B}$ is turned on ($V_{C_P} < V_{C_1}$).

In this case, the voltage on the piezoelectric film is:

$$V_P = V_{C_1} = \left(1 + \frac{1}{4}\right) \frac{C_1 C_P}{(C_P + C_1)^2} \left(V_{D_1} + V_{D_2} + V_R\right) \tag{24}$$

Simplification can be obtained:

$$V_P = V_{C_1} = \frac{5}{16} \left(V_{D_1} + V_{D_2} + V_R\right) \tag{25}$$

The current equation for the SSHC circuit is:

$$I_P(t) + C_p \frac{d(V_P)}{dt} + \frac{V_P}{R_P} = C_1 \frac{dV_{C_1}}{dt} \tag{26}$$

The SSHC circuit completes the flip from negative maximum to positive voltage; during this time $V_P > 0$. Combined with the absolute values of two voltage flips, it should be noted that the flipped voltage shown in Figure 14 is higher than the flipped voltage in Figure 10 which means more charge is transferred in this process.

Then, the piezoelectric film again moves toward the maximum forward displacement. Current flow of the SSHC circuit is shown in Figure 7. Current and voltage on the piezoelectric flag again begin to grow negatively. The current and voltage repeat the steps discussed above.

The reversal of the voltage on the piezoelectric film is accumulated in every cycle. The reversal of the voltage at both ends of the piezoelectric element at each current zero-crossing time is accumulated, referred to as N cycles of voltage reversal. After the first reversal of voltage:

$$|V_P| = V_{C_1} = \frac{1}{4}\left(V_{D_1} + V_{D_2} + V_R\right) \tag{27}$$

After the second reversal of voltage:

$$|V_P| = V_{C_1} = \left(\left(\frac{1}{4}\right)^2 + \frac{1}{4}\right)\left(V_{D_1} + V_{D_2} + V_R\right) \tag{28}$$

After the $N$th voltage reversal:

$$|V_P| = V_{C_1} = \left(\left(\frac{1}{4}\right)^n + \cdots + \left(\frac{1}{4}\right)^2 + \frac{1}{4}\right)\left(V_{D_1} + V_{D_2} + V_R\right) \tag{29}$$

The limit for Formula (29) is:

$$\lim_{n \to \infty} |V_P| = \frac{1}{3}\left(V_{D_1} + V_{D_2} + V_R\right) \tag{30}$$

Under the condition that the clamping capacitor $C_P$ on the piezoelectric film is equal to switchig capacitor $C_1$, the flip efficiency of the SSHC circuit can reach 1/3. The theoretical voltage waveform with the SSHC circuit is shown in Figure 15.

Assuming that the components in the circuit are ideal elements, during time interval $[t_0, t_1]$, switch $S_{1A/B}$ is turned on and electric current flows into switching capacitor $C_1$. During $[t_1, t_2]$, the piezoelectric film is in the short-circuit state. During time interval $[t_2, t_3]$, the switching capacitor $C_1$ charges the piezoelectric film in the opposite side through switches $S_{3A}$, $S_{3B}$. According to the law of conservation of charge, the current flowing out of the piezoelectric flag can be expressed as:

$$\left|\int_{t_0}^{t_0+\frac{T}{2}} I dt\right| = \int_{t_0}^{t_1} i_{C_1} dt + \int_{t_1}^{t_2} i_{R_0} dt - \int_{t_2}^{t_3} i_{C_0} dt + \left|\int_{t_3}^{t_3+\frac{T}{2}} i_R dt\right| \tag{31}$$

By substituting Equation (3) into Equation (31), the left side of Equation (31) can be simplified to:

$$\int_{t_0}^{t_0+\frac{T}{2}} I dt = \int_{t_0}^{t_0+\frac{T}{2}} \left(\alpha \dot{u} - C_0 \dot{V}_{C0}\right) dt = -2(\alpha U_M - C_0 V_{DC}) \tag{32}$$

Within $[t_0, t_1]$, the voltage reversal coefficient is set as $\lambda = V_{C_P}/V_M$ and the following relationship can be obtained:

$$\int_{t_0}^{t_1} i_{C_1} dt = \lambda C_1 V_R \tag{33}$$

Within $[t_1, t_2]$:

$$\int_{t_2}^{t_1} i_{R_0} dt = \frac{(1-\lambda)V_M}{R_0} \tag{34}$$

Within $[t_2, t_3]$:

$$\int_{t_2}^{t_3} i_{C_0} dt = \lambda C_0 V_R \tag{35}$$

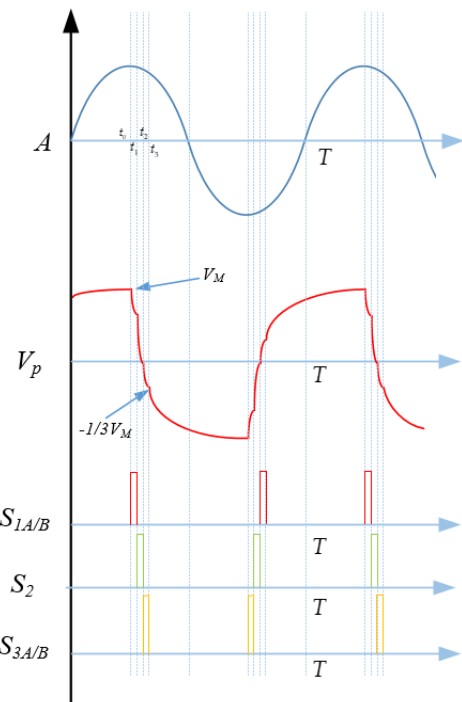

**Figure 15.** Theoretical waveform of the SSHC circuit.

Assuming that the voltage $V_R$ of the load remains constant within $[t_2, t_2 + T/2]$, the following relationship can be obtained:

$$\int_{t_2}^{t_2+T/2} i_R dt = \frac{V_R}{R} \cdot \frac{T}{2} \tag{36}$$

Inserting Equations (32)–(36) into Equation (31):

$$V_R = \frac{2R\alpha\omega U_M}{(1-\lambda)\omega R C_0 + \pi} \tag{37}$$

Thus, the output power is:

$$P_{SSHC} = \frac{V_R}{R} = \frac{4R^2\alpha^2\omega^2 U_M^2}{[(1-\lambda)\omega R C_0 + \pi]^2} \tag{38}$$

From Equation (38), the output power of the SSHC circuit can be regarded as a function of resistance $R$. The optimal load of the circuit from $(dP_{SSHC})/dR = 0$ can be derived:

$$R_{opt} = \frac{\pi}{2(1-\lambda)\omega C_P} \tag{39}$$

The corresponding maximum output power is:

$$P_{SSHC-M} = \frac{\omega\alpha^2 U_M^2}{2(1-\lambda)\pi C_P} \tag{40}$$

### 3.3. Switching Delay Time of the SSHC Circuit

The switch action in the circuit occurs at the time when the current passes zero, and the control system has a certain reaction delay time from detecting zero current to sending the control signal to make the switch on. In order to ensure that the electric energy is not wasted and the voltage switching efficiency of the circuit is ensured, it is necessary to analyze the corresponding response delay time when the switch starts to operate.

The zero-crossing moment when the current reaches from positive to negative is taken as an example, assuming that $I_P(t) = \cos t$ is the current generated by the piezoelectric flag; $\Delta t$ represents the delay time from the detection of the current to the time the switch turns on. The on-time of the switch is $\tau$. The relationship between amplitude, circuit current, and terminal voltage of the piezoelectric film is shown in Figure 16.

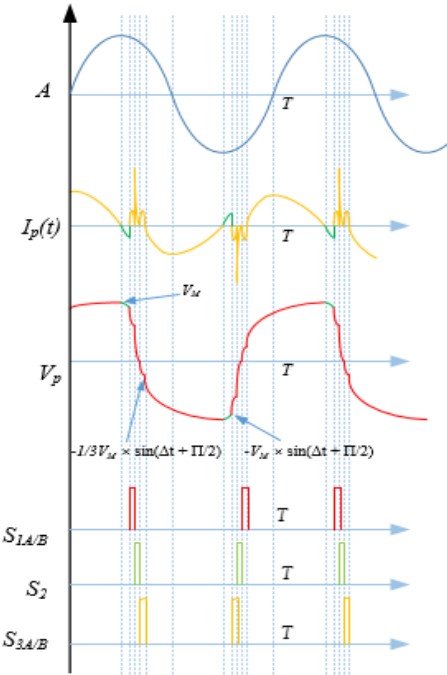

**Figure 16.** Theoretical waveform of the piezoelectric flag with delay.

At this moment, the current $I_P$ from the positive to negative side reaches the zero-crossing point, and the terminal voltage of the piezoelectric flag decreases slightly from its maximum value along the sine curve. After delay $\Delta t$:

$$V_P = V_M \times \sin\left(\Delta t + \frac{\pi}{2}\right) \tag{41}$$

Switches $S_{1A/B}$, $S_2$, $S_{3A/B}$ are then turned on in sequence and the sum of on-time is $3\tau$. The terminal voltage of the piezoelectric flag after flipping is:

$$V_P = -\frac{1}{3}V_M \times \sin\left(\Delta t + \frac{\pi}{2}\right) \tag{42}$$

At time $(\Delta t + 3\tau)$ past when the current crosses the zero-point, the piezoelectric film begins to move to the negative maximum displacement, and the voltage and current begin to decrease to the negative maximum.

If the delay time $\Delta t$ is very small, the voltage on the piezoelectric film reaches its maximum before the next time interval $(3\pi/2) - (\Delta t + 3\tau)$. If the delay time $\Delta t$ is too large, then the voltage of the piezoelectric film cannot reach its maximum before the next time interval $(3\pi/2) - (\Delta t + 3\tau)$.

From the time of voltage flipping to the next zero-crossing moment of current, the voltage increment is:

$$\Delta V = \frac{1}{C} \int_{\frac{3\pi}{2}+\Delta t+3\tau}^{\frac{3\pi}{2}} i\,dt \tag{43}$$

Substituting $I_P(t) = \cos t$ and solving:

$$\Delta V = -\frac{1}{C}[1 + \cos(\Delta t + 3\tau)] \tag{44}$$

The sum of the voltage value and voltage increment after flipping is no less than its original maximum voltage value:

$$-\frac{1}{3}V_M \times \sin\left(\Delta t + \frac{\pi}{2}\right) - \frac{1}{C}[1 + \cos(\Delta t + 3\tau)] \geq -V_M \tag{45}$$

The delay time $\Delta t$ can be obtained by solving the inequality:

$$\Delta t \leq \arcsin\left\{ \frac{-\left(V_M - \frac{1}{C}\right)\left(\frac{1}{C}\sin 3\tau\right) + \sqrt{\begin{array}{c} 2\left(V_M - \frac{1}{C}\right)^2 \times \left(\frac{1}{C}\sin 3\tau\right)^2 \\ -2\left[\left(\frac{1}{3}V_M + \cos 3\tau\right)^2 + \left(\frac{1}{C}\sin 3\tau\right)^2\right] \\ \times\left[\left(V_M - \frac{1}{C}\right)^2 - \left(\frac{1}{3}V_M + \cos 3\tau\right)^2\right] \end{array}}}{\left(\frac{1}{3}V_M + \cos 3\tau\right)^2 + \left(\frac{1}{C}\sin 3\tau\right)^2} \right\} \tag{46}$$

Above all, the delay time $\Delta t$ cannot be unavoidable in the control of the SSHC circuit because of system response and switch delay, but there is a critical value $\Delta T$. If $\Delta t < \Delta T$, the voltage on the piezoelectric film can reach its maximum in the time that the current crosses the zero-point. The electric energy is not wasted and efficiency with the SSHC is improved.

*3.4. Simulation of Interface Circuit*

A simulation with Multisim software 14.1 was performed to compare the SEH and SSHC interface circuits for the piezoelectric flag. Because the equivalent resistor of PVDF $R_p$ is very large, it can be ignored or considered as an open circuit. A current source paralleled with a capacitor was proposed to model the piezoelectric flag, as shown in Figure 17. A rectifier of four BAT86 diodes converts alternating current to direct current. Capacitor $C_5$ stabilizes the voltage to power the load $R_2$, and the voltage waveform of the piezoelectric flag is shown in Figure 18.

Figure 19 shows the voltage waveform of load in the SEH circuit. Load voltage fluctuates around 2.4 V. Because of the continuous consumption of load, the energy stored by capacitance $C_{23}$ is limited, the voltage drops rapidly, and the short cycle of AC can quickly boost the capacitor, so the voltage fluctuates up and down.

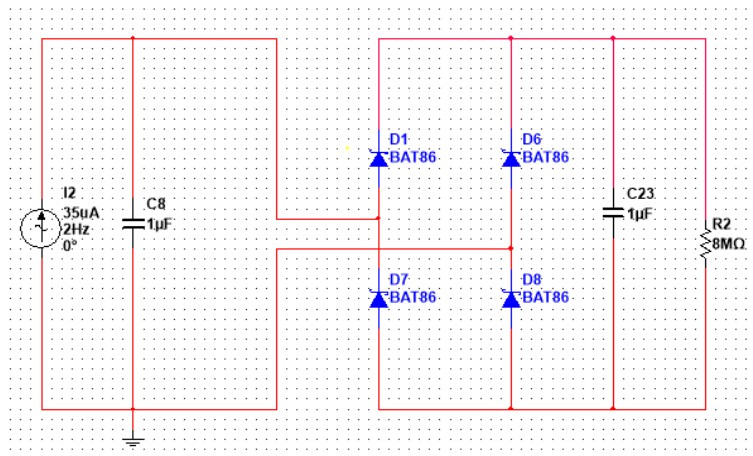

**Figure 17.** Simulation of SEH in Multisim software.

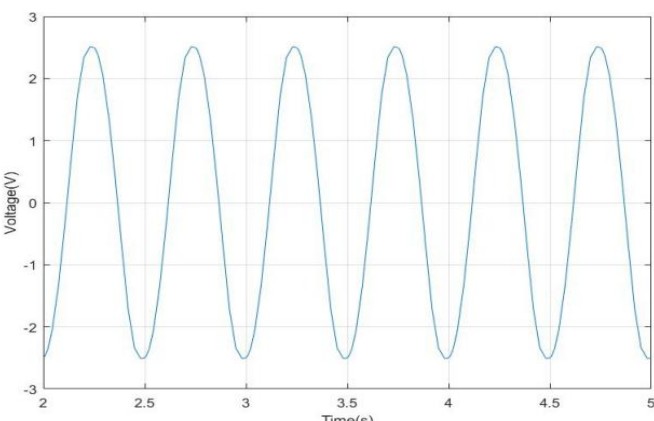

**Figure 18.** Voltage waveform of the piezoelectric flag with SEH.

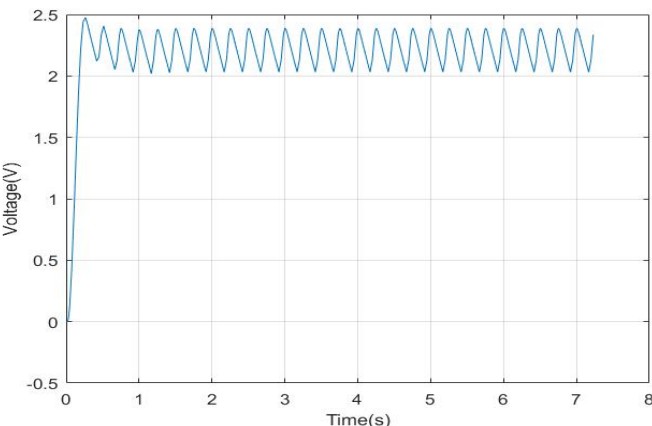

**Figure 19.** Voltage waveform of load in the SEH circuit.

Instead of an inductor, the SSHC interface circuit can flip the voltage across the piezoelectric film with capacitors and switches. To simplify the simulation program, each switch in Figure 6 was modeled by two switches, and these switches are controlled by six pulse signals. The pulse sources $V_1$, $V_2$, $V_3$. control the switches when the current $I_p$ crosses the zero-point from positive to negative. The pulse sources $V_4$, $V_5$, $V_6$ control the switches when the current $I_p$ crosses the zero-point from negative to positive, as shown in Figure 20. The simulation waveform of the piezoelectric films is shown in Figure 21.

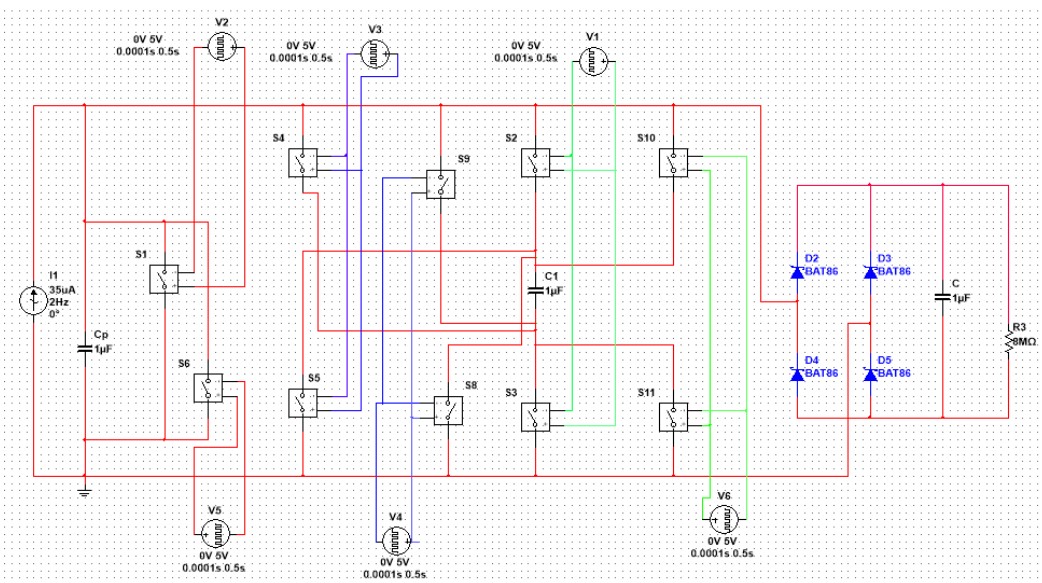

**Figure 20.** Simulation of the SSHC in Multisim software.

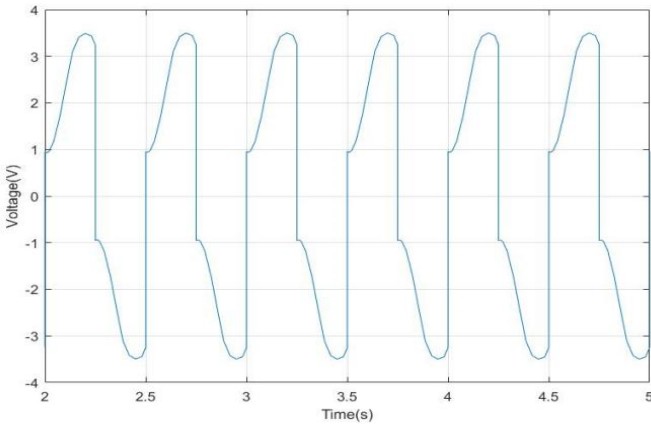

**Figure 21.** Simulation waveform of the SSHC circuit.

For each zero-crossing moment, $V_P$ goes to an opposite value and a part of the charge is inverted by switches and capacitor $C_1$. At the instant of 3 seconds, the current of the piezoelectric films crosses the zero-point and switches $S_{1A}$, $S_{1B}$ turn on. Some charge on the piezoelectric films flows to capacitor $C_1$, which leads to a voltage drop of the piezoelectric flag. Then, switches $S_{1A}$, $S_{1B}$ turn off and switch $S_2$ turns on. As a result of the short circuit, the remaining charge on the piezoelectric films is cleared and its voltage achieves zero. Next, switch $S_2$ turns off and switches $S_{3A}$, $S_{3B}$ turn on. Some charge on $C_1$ flows back to the piezoelectric films in an opposite side and it leads to a reversed rising voltage of the piezoelectric flag. Details of voltage flip are shown in Figures 22 and 23; the voltage flip efficiency can achieve $1/3$ in the case of $C_P = C_1$ [16].

The voltage waveform of load in the SSHC circuit is shown in Figure 24. When the waveform is stable, the voltage waveform is sinusoidal. The voltage fluctuates up and down at 3 V. This is because with every switching action, the piezoelectric flag is connected with the switched capacitor $C_1$; the next moment it is shorted and reverse voltage applied. During this series of actions, the voltage of the load is consumed, and when the voltage is reversed, the load increases rapidly. The load voltage presents a sine wave since the switch action is completed in a very short time.

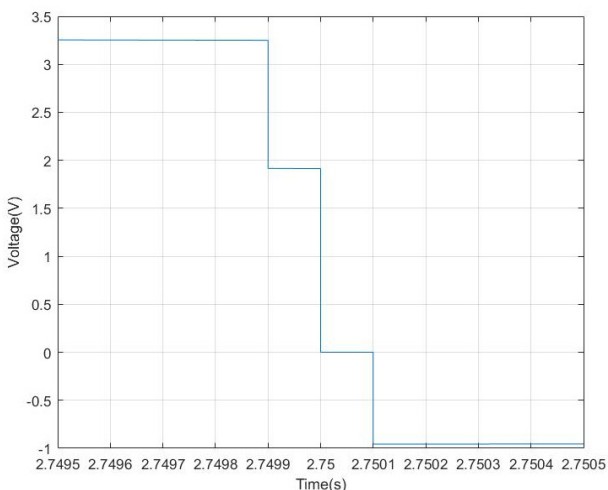

**Figure 22.** Zero-crossing moment while $V_P$ is inverted from "+" to "−".

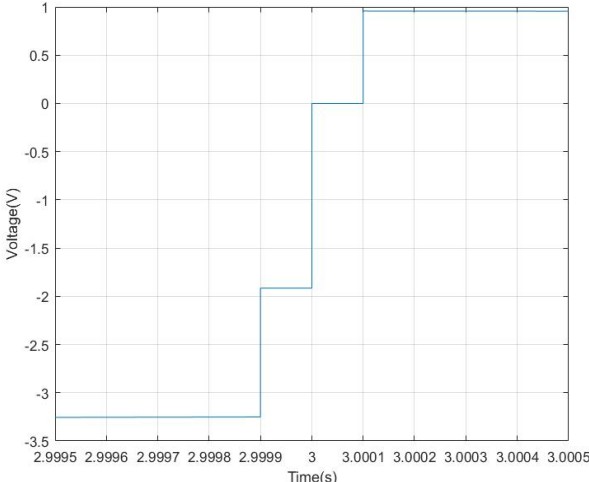

**Figure 23.** Zero-crossing moment while $V_P$ is inverted from "−" to "+".

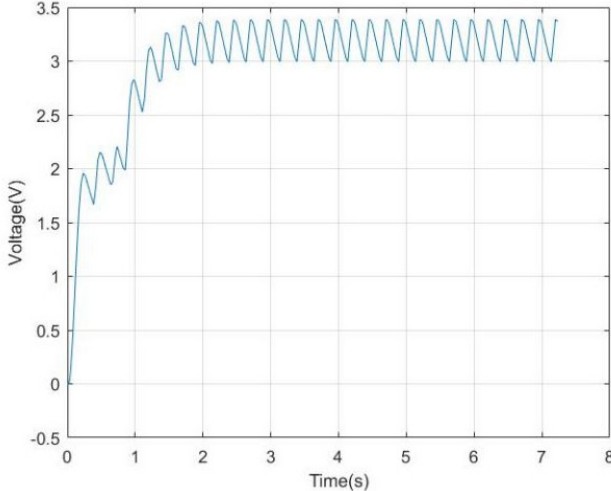

**Figure 24.** Voltage waveform of load in the SSHC circuit.

To compare the SEH and SSHC interface circuits, different load resistances were connected to their output and the output power measured. The output power with SSHC is higher than that with SEH at resistances from 500 to 2000 kΩ. With SSHC, 44% more power can be obtained than with SEH at a load of 1700 kΩ, as shown in Figure 25.

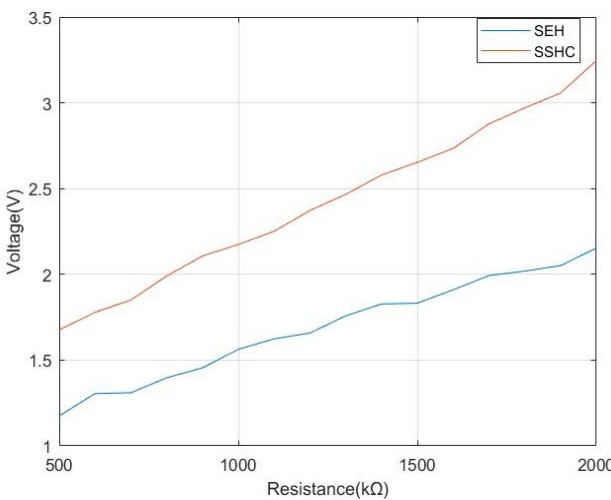

**Figure 25.** Voltage–resistance diagram for SEH and SSHC circuits.

The on/off switch is controlled by six pulsating voltage sources. The pass time of the switch is set to 0.001 s, and the switch turn-on period is 0.5 s. From switch $S_{1A}$ to $S_{3B}$, the delay times of switch-on are in the sequence 0.25, 0.251, 0.252, 0.5, 0.501, and 0.502 s. The SEH and SSHC circuits are placed in the same simulation interface. The capacitance value of the switching capacitance and the clamping capacitance of the two circuits are set to the same value. The simulation waveforms of the SEH and SSHC circuits are shown in Figure 26.

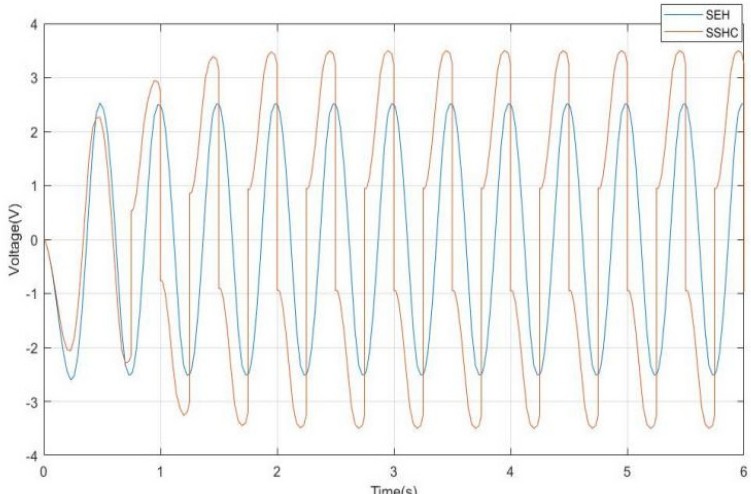

**Figure 26.** Simulation waveforms of the SEH and SSHC circuits.

After the waveform is stable, the piezoelectric flag voltage of the SSHC circuit is higher than that of the SEH circuit. The voltage of the SSHC circuit is about 1 V higher than the voltage of the SEH circuit voltage.

## 4. Experiment

### 4.1. Experiment Setup

An experiment platform was built to test the proposed interface circuits, as shown in Figure 27. To test the proposed SSHC interface circuit, a small piezoelectric flag was designed with a resonant frequency higher than the larger flags. If the proposed SSHC interface circuit works well at high frequency and high wind speed, then it will certainly work well at low frequency and low wind speed for larger piezoelectric flags.

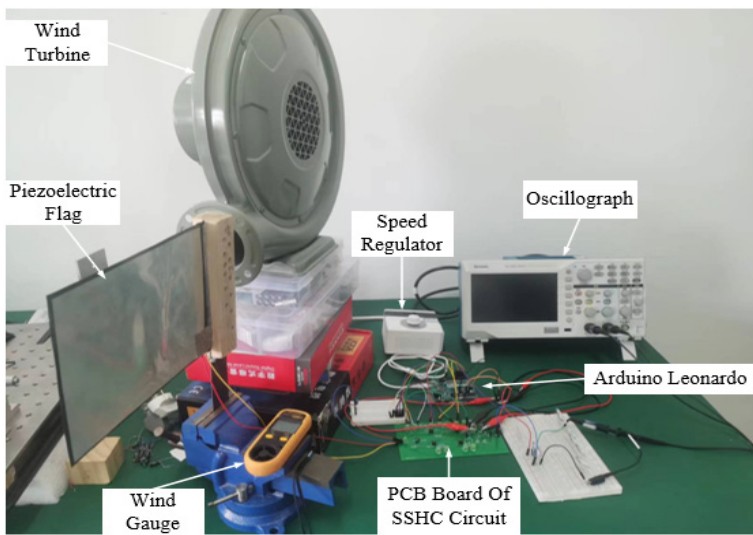

**Figure 27.** Experiment setup.

The wind from the wind turbine was adjusted by a speed regulator and measured by a wind gauge. The piezoelectric films were bonded on a steel plate which was fixed on a blunt body of wood. A PCB board was designed for the SSHC interface circuit. An Arduino Leonardo microcontroller was used to run the control algorithm for the SSHC. The voltage on the piezoelectric films and the output voltage were measured and monitored by a multimeter (Fluke 54564654) and an oscilloscope (Tektronix TBS1072C).

### 4.2. Interface Circuit Design and Control

As shown in Figure 28, a PCB board was design to connect the piezoelectric film and load. Because the pulse signals from the microcontroller are common grounded, five optical couplers are necessary to perform as switches, as shown in Figure 6. These five optical couplers of type APY212RE(DIP-4) are controlled by three pulse signals from the Arduino Leonardo microcontroller. A 1 µF capacitor with the five optical couplers and microcontroller can flip the voltage to perform SSHC. Four diodes (BAT86) and a 1 µF capacitor rectify the voltage and power the load. The electric components are listed in Table 1.

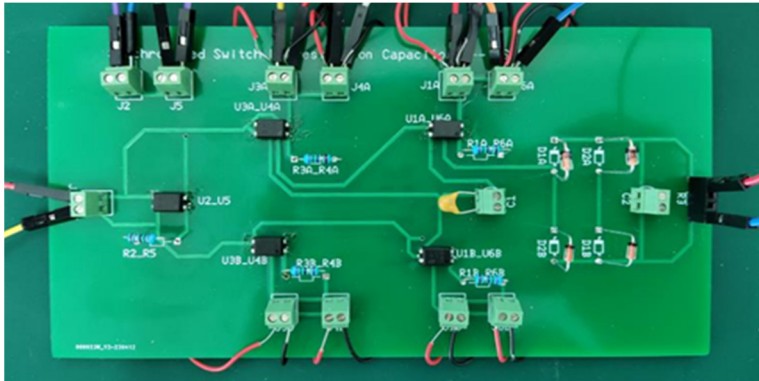

**Figure 28.** SSHC interface circuit.

**Table 1.** Electronic components.

| Device | Type | Number |
| --- | --- | --- |
| Optical coupler | APY212RE(DIP-4) | 5 |
| Capacitor | 1 µF | 2 |
| Diodes | BAT86 | 4 |

As presented in Section 3.2, switch $S_{1A}$, $S_{1B}$ turns on at the time that current $I_p$ from the positive to negative side reaches the zero-crossing point. In real-time control, it is difficult to monitor the current through the piezoelectric films with less power consumption. A method of monitoring diode voltage was proposed to replace current monitoring. If the negative end of $V_R$ was chosen as the ground of the microcontroller, then the voltage on diode $D_1$, $D_4$ changes with the current $I_p$. When the current $I_p$ reaches the zero-crossing point from positive, then the voltage on $D_1$ changes from positive to negative and the voltage on $D_4$ changes from negative to positive. If the current $I_p$ reaches the zero-crossing point from the negative, then the voltage on $D_1$ changes from negative to positive and the voltage on $D_4$. changes from positive to negative.

As presented in Section 3.3, a control algorithm was designed to detect the voltage and control the switches. The control algorithm is programmed and operated in the Arduino Leonardo microcontroller, as shown in Figure 29. The voltages on $D_1$, $D_4$ are detected by the sampling ports on the Arduino Leonardo. Switches are controlled by the output ports on the Arduino Leonardo. Software flow charts for the proposed algorithm are shown in Figure 30 and Appendices A and B.

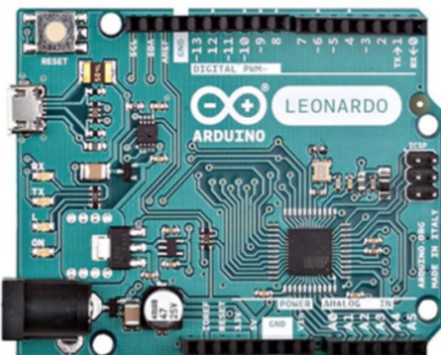

**Figure 29.** Arduino Leonardo microcontroller.

After the system initialization, global variables were defined and assigned their initial value. Analog port A0, A1 was configured as the input port and digital port D3, D4, D5, D6, D7, D8 was configured as the output port. Port A0 acquires the voltage on diode $D_1$ and Port A1 acquires the voltage on diode $D_4$. These two voltages were compared by $V_0 - V_1$, and their difference was assigned to variable Diff_0. Subroutines f1 and f2 were then called in order. Finally, the value of Diff_0 was assigned to Diff_p1 and the loop ended.

In subroutine f1, if variable Diff_p1 is larger than R1 and Diff_0 is smaller than R1, then value 6 is assigned to the global variable Flag1 and the pulse signal is sent out by the Arduino Leonardo board. By setting the initial value of R1, the pulse time can be adjusted to control the $S_{1A/B}$, $S_2$ and $S_{3A/B}$ switches.

If variable Diff_p1 is smaller than R1 or Diff_0 is larger than R1, then switch $S_{1A/B}$ is turned on in the case of Flag1 > R1_3. In this algorithm, the intersection of voltage V0 and V1 can be found by the difference of the acquired voltage, and the on-time of switch $S_{1A/B}$ is adjusted by the values of Flag1 and R1_3. If Flag1 $\leq$ R1_3 and Flag1 > R1_4, then switch $S_2$ is turned on and its on-time is controlled by the values of R1_3 and R1_4. In the case of Flag1 $\leq$ R1_4 and Flag1 > R1_5, then switch $S_{3A/B}$ is turned on and its on-time is adjusted by configuring the variables R1_4 and R1_5.

The last step of Subroutine f1 is that the value of Flag1-1 is assigned to variable Flag1. After Subroutine f1 ends, the subprogram f2 is called, and its process is similar to the process used in Subroutine f1. With this algorithm process, the switches are controlled to turn on or off when current $I_P$ reaches the zero-crossing point.

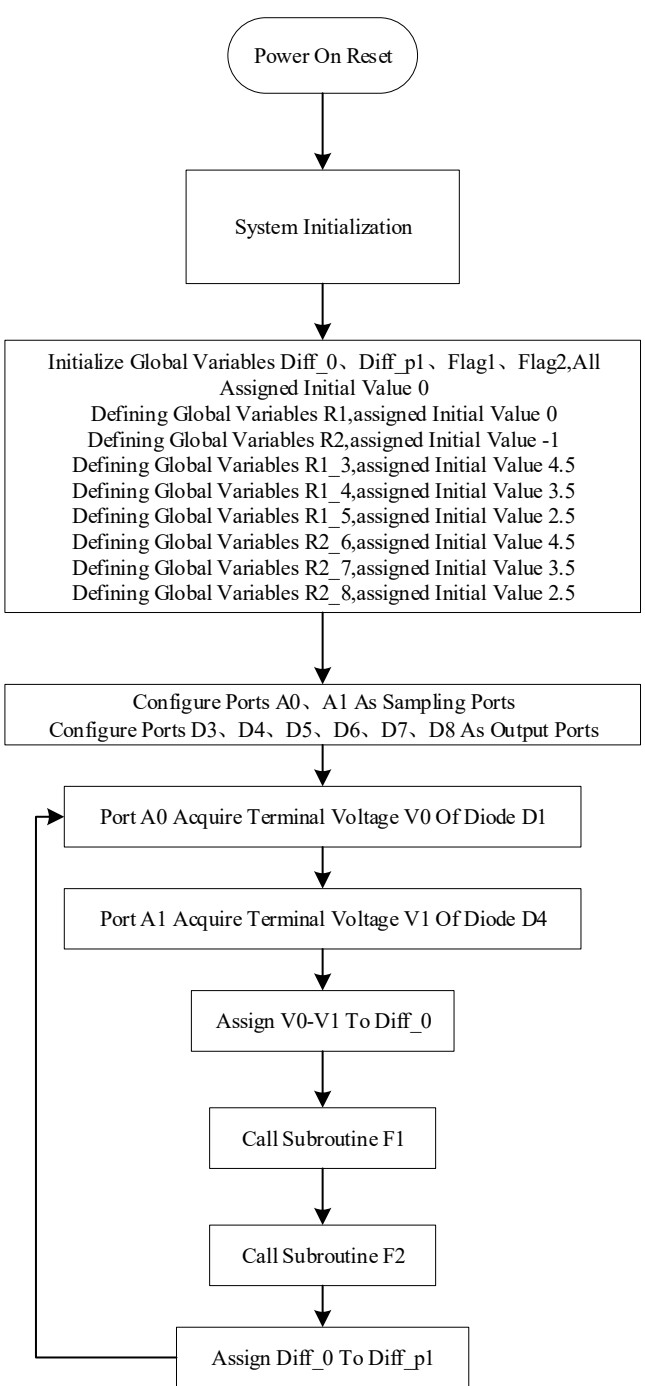

**Figure 30.** Flow chart of the control algorithm.

*4.3. Results and Discussion*

With the designed PCB board and proposed control algorithm, experiments were carried out using the devices shown in Figure 27. The voltages on the piezoelectric films with the SEH and SSHC interface circuit were monitored by oscilloscope, as shown in Figure 27. Comparing Figures 31 and 32, it can be found that the voltage on the piezoelectric films with SSHC is flipped and about 28% higher than voltage with SEH. This result indicates that the designed PCB board and the proposed control algorithm work properly.

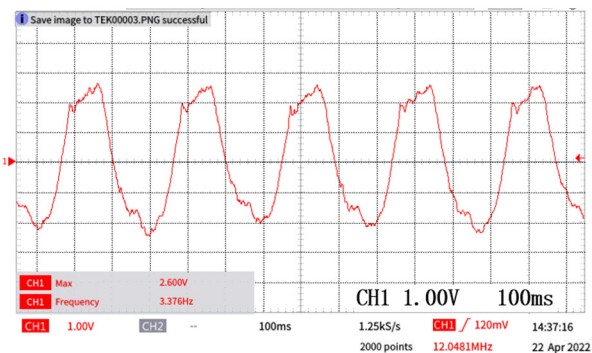

**Figure 31.** Voltage on the piezoelectric flag with SEH.

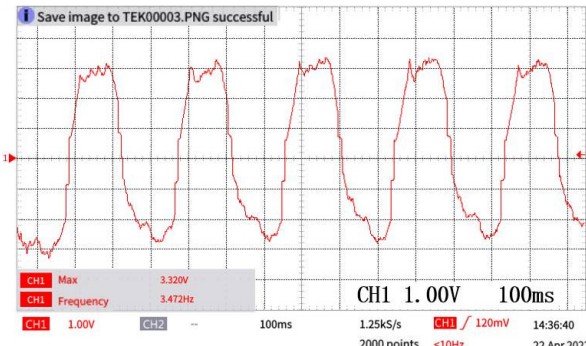

**Figure 32.** Voltage on the piezoelectric flag with SSHC.

To compare the output voltage and power with different interface circuits, the piezoelectric flag is first tested with the SEH interface circuits. With the load resistance from 500 to 2000 kΩ, the output voltage on load was monitored by oscilloscope. As shown in Figure 33, the output voltage with SEH increases with load resistance at a fixed wind speed. The maximum voltage at a wind speed of 18 m/s is 2.064 V and it achieves 2.74 V at 24 m/s. Similar to SEH, the output voltage with SSHC also increases with load resistance, as shown in Figure 34. It should be noted that the output voltage with SSHC is about 45% higher than the output voltage with SEH at all wind speeds and load resistances.

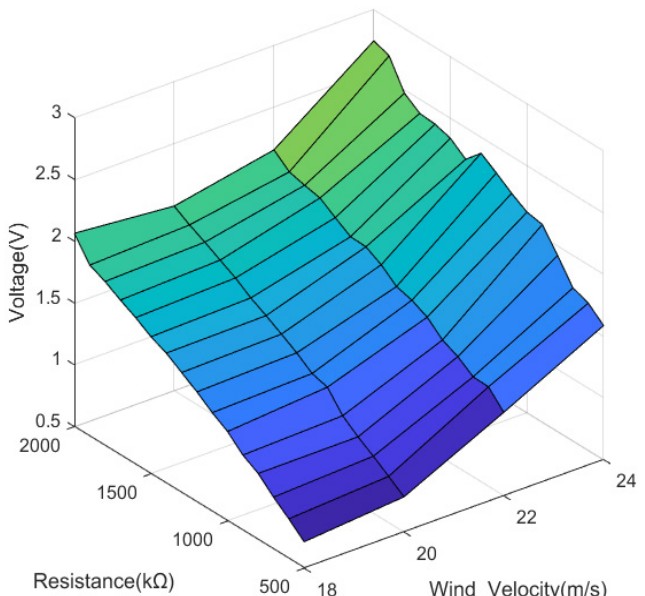

**Figure 33.** Output voltage with SEH for different loads and wind speeds.

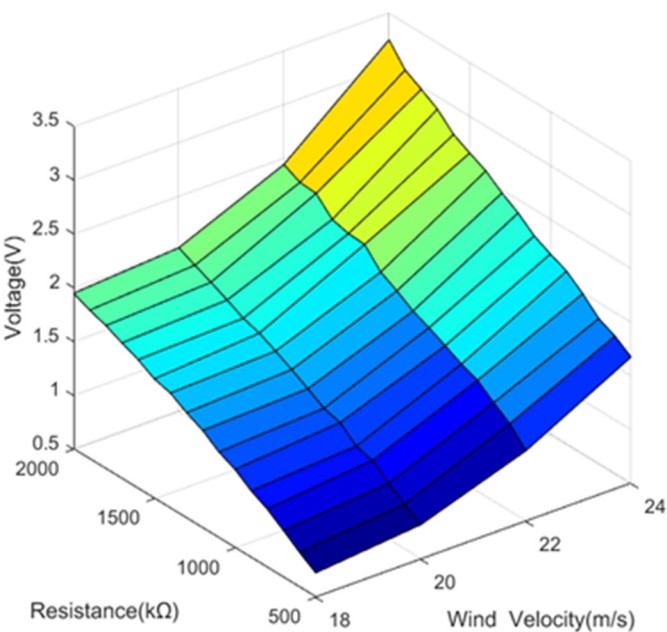

**Figure 34.** Output voltage with SSHC for different loads and wind speeds.

At the wind speed of 18 m/s, the relationship between output power and resistance of the SEH and SSHC circuits is shown in Figure 35. Observe that the output power of the two circuits increases with the increase in resistance, but the difference between the two is not large, which is due to the low vibration amplitude of the piezoelectric flag structure and the small output power under this wind speed. When the load resistance is 1400 kΩ, the output power of SSHC can be increased by 2.4% compared with the SEH circuit.

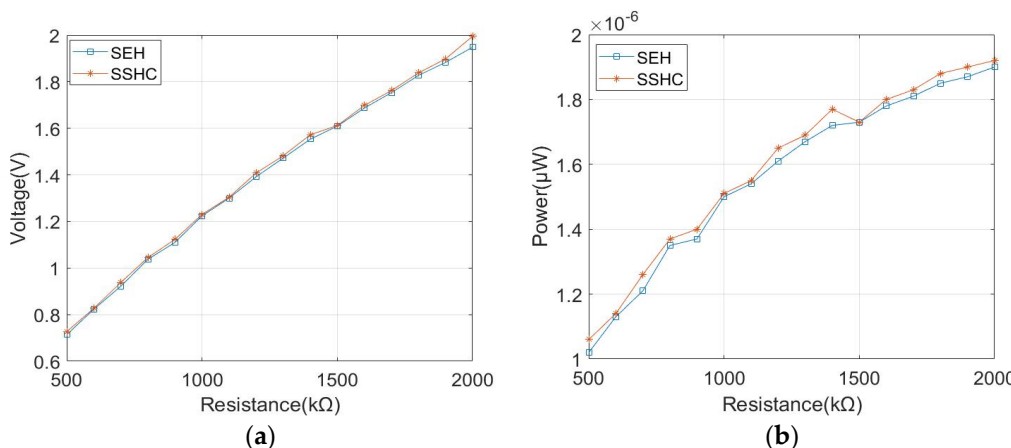

**Figure 35.** Output voltage (**a**) and power (**b**) of the SSHC circuit at a wind speed of 18 m/s.

At a wind speed of 20 m/s, the relationship between output power and resistance of the SEH and SSHC circuits is shown in Figure 36. Output power of both circuits increases with the increase in resistance. The difference of output power between the SEH and SSHC circuits increases with the increase in vibration amplitude of the piezoelectric flag. When the resistance is 800 kΩ, the output power of SSHC can be increased by 20.98% compared with the SEH circuit.

At a wind speed of 22 m/s, the relationship between output power and resistance of the SEH and SSHC circuits is shown in Figure 37. The output power of SEH circuit decreases with the increase in load resistance. Output power of the SSHC circuit fluctuates around 2.9 μW. When the resistance is 1500 kΩ, the output power of SSHC can be increased by 38.49% compared with the SEH circuit.

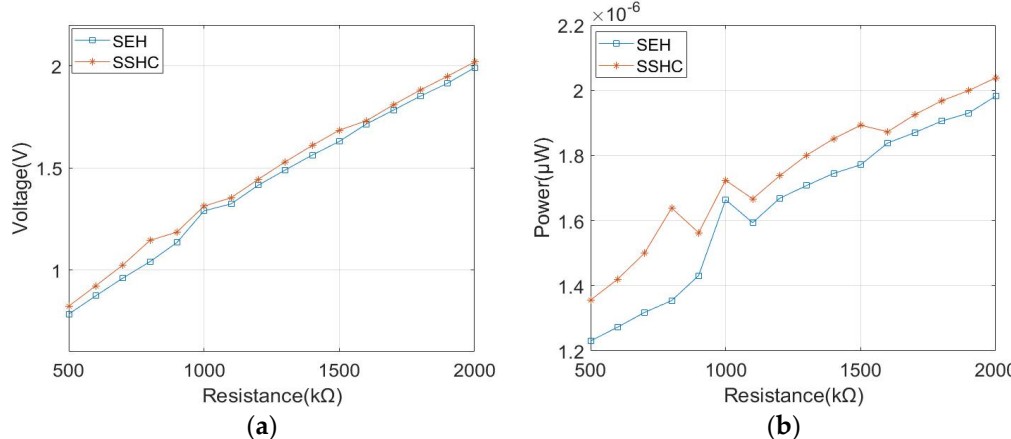

**Figure 36.** Output voltage (**a**) and power (**b**) of the SSHC circuit at a wind speed of 20 m/s.

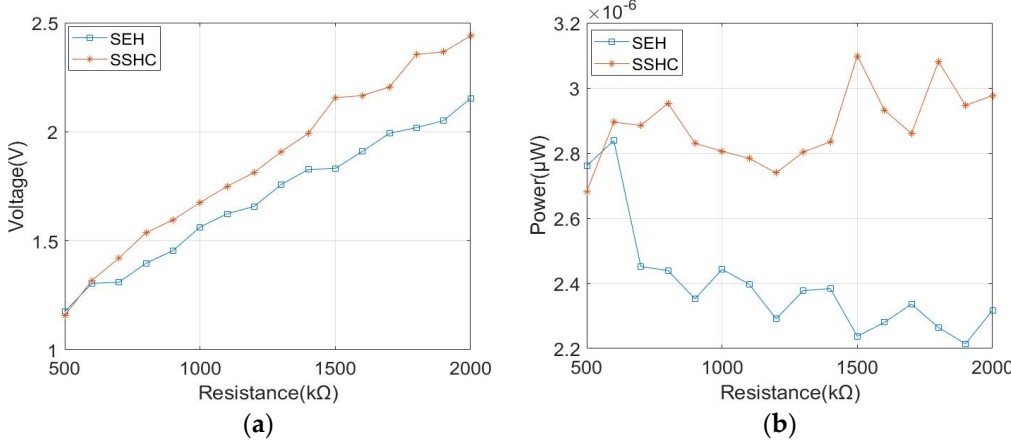

**Figure 37.** Output voltage (**a**) and power (**b**) of the SSHC circuit at a wind speed of 22 m/s.

At a wind speed of 24 m/s, the relationship between output power and resistance of the SEH and SSHC circuits is shown in Figure 38. It can be seen that the output power of the SEH circuit decreases with the increase in load resistance. The output power of the SSHC circuit decreases first and then increases with the increase in resistance. When the resistance is 1700 kΩ, the output power of SSHC can be increased by 45.63% compared with the SEH circuit.

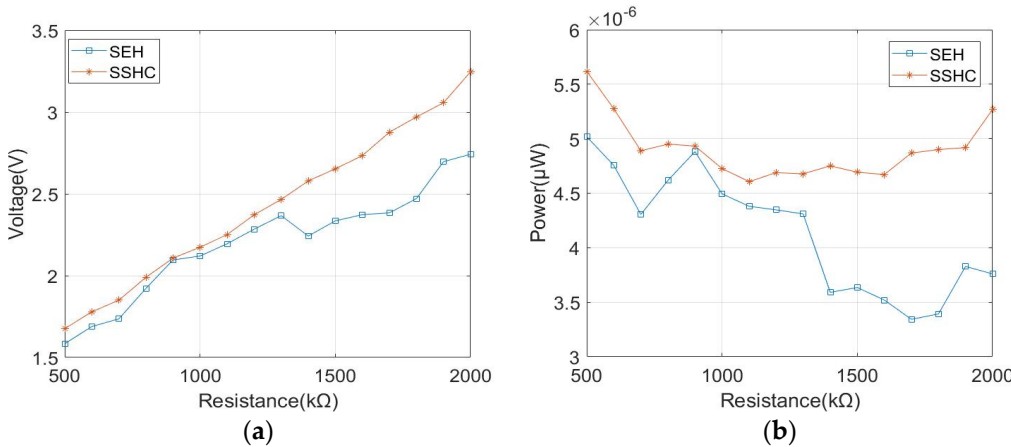

**Figure 38.** Output voltage (**a**) and power (**b**) of the SSHC circuit at wind speed of 24 m/s.

## 5. Conclusions

The interface circuits for energy harvesting of a piezoelectric flag were analyzed, and it was found that only SEH and SSHC are suitable for piezoelectric films. Simulation using Multisim software was performed to compare the SEH and SSHC circuits under different load resistance. A PCB board was designed and a control algorithm was developed on a microcontroller. Experiments were carried out using different wind speeds and load resistances. The results indicate that the output voltage with SSHC is higher than the output voltage with SEH at all wind speeds and load resistances.

**Author Contributions:** Conceptualization, Y.C.; methodology, Y.C., L.T. and P.Z.; software, Y.C. and P.Z.; validation, Y.C., L.T. and P.Z.; formal analysis, Y.C.; L.T. and P.Z.; investigation, J.D.; resources, H.J.; data curation, L.T.; writing—original draft preparation, Y.C.; writing—review and editing, Y.C., L.T.; visualization, H.J.; supervision, H.J.; project administration, J.D.; funding acquisition, H.J. All authors have read and agreed to the published version of the manuscript.

**Funding:** The work was financially supported by funding for school-level research projects of the Yancheng Institute of Technology (No. xjr2021016, xj201528), the National Natural Science Foundation of China (No. 52022039).

**Institutional Review Board Statement:** Not applicable.

**Informed Consent Statement:** Not applicable.

**Data Availability Statement:** Not applicable.

**Acknowledgments:** The author sincerely thanks Yancheng Institute of technology, Nanjing Institute of technology and Nanjing University of Aeronautics and Astronautics for their support. The author sincerely thanks the reviewers for their valuable comments, which has substantially improved the paper.

**Conflicts of Interest:** The authors declare no conflict of interest.

## Appendix A

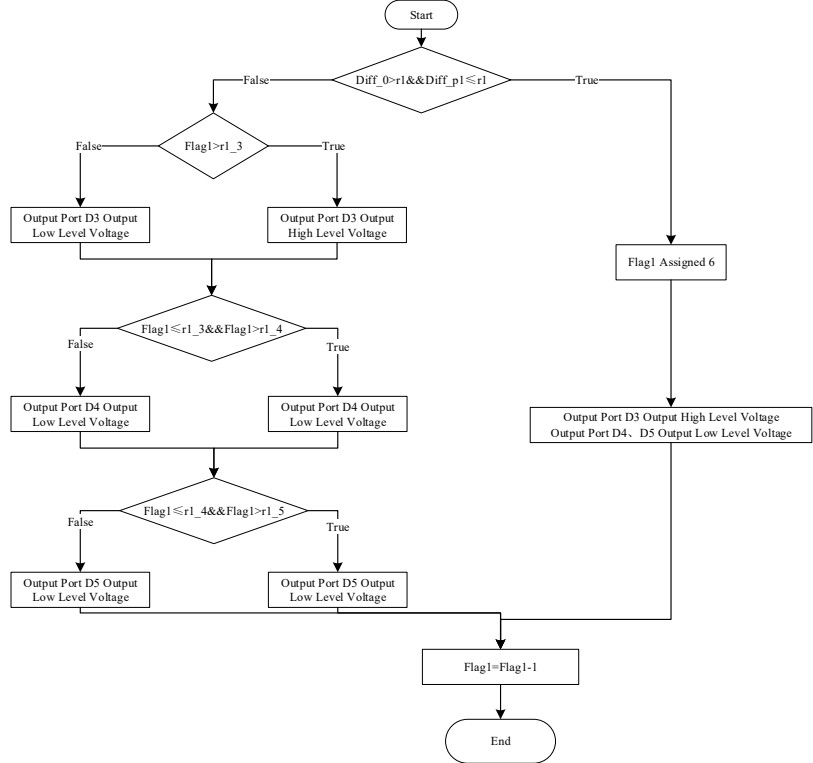

**Figure A1.** Flow chart of Subroutine f1.

## Appendix B

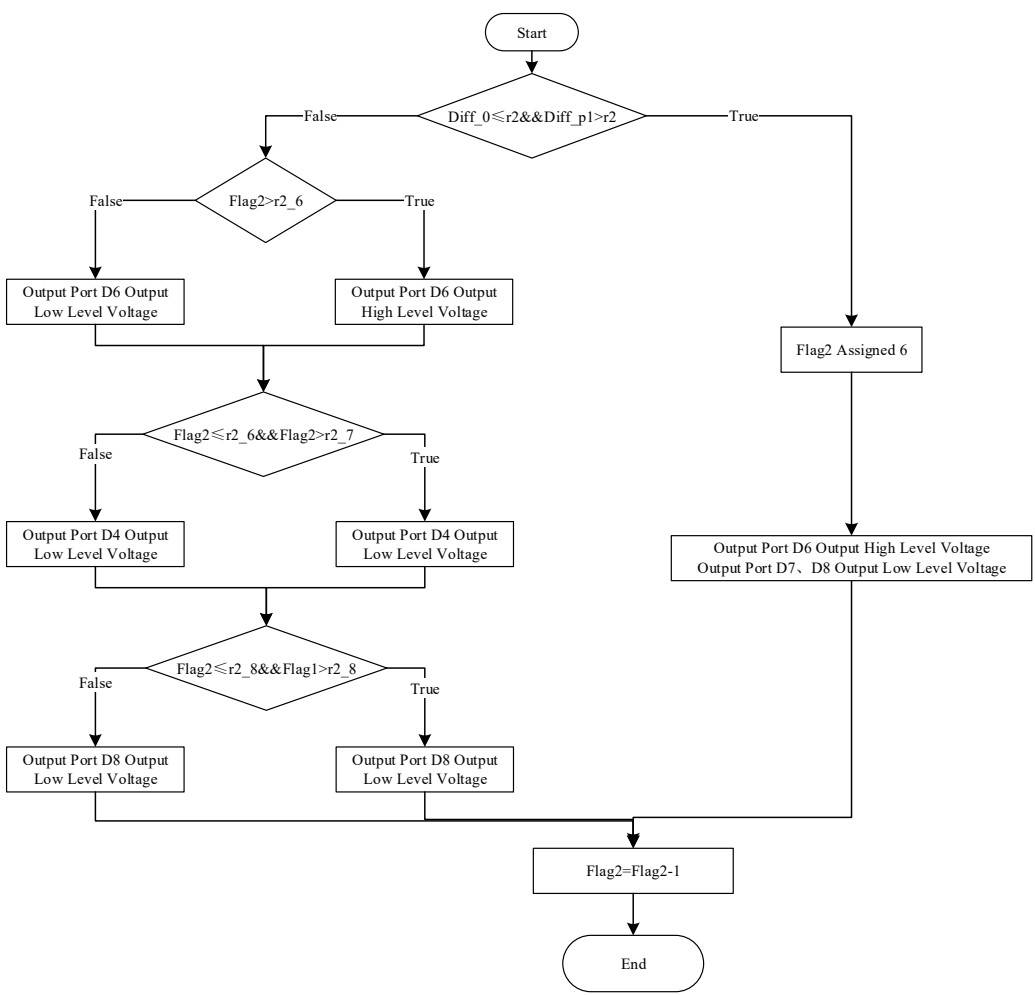

**Figure A2.** Flow chart of Subroutine f2.

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
