# Peer review of "An SSHC Interface Circuit for Energy Harvesting of Piezoelectric Flags"

_actuators, doi:10.3390/act11070175_

Round 1

Reviewer 1 Report

A novel structure of piezoelectric flag was designed to generate the vibration by wind inducing. The effectiveness and advantages of the proposed structure are verified by the simulation and experiment results. The paper is interested and can be accepted after revised the following comments.

1.        The cited number of reference paper should be in order. The number [5] should be after the number [4] in Page 1.

2.        It should be a mistake of reference [1] or [5]. These two papers should not be the same.

3.        Why the output voltage is increasing in Figure 34 and 35, but the output power is decreasing in Figure 38 and Figure 39.

4.        The PVDF piezoelectric film is equivalent to a current source, capacitor and resistor in Figure 8-15, but it is only equivalent to a current source and capacitor in Figure 18 and 21. Why?

5.        In Page 11, the time interval [t2, t3] is not discussed. A detailed description is suggested.

6.        In Figure 7, there are only 5 switches in the diagram of SSHC interface circuit, but there are 10 switches in the simulation in Multisim. Why?

7.        There is an elastic steel in the structure of piezoelectric flag in Figure 3. Why it is not mentioned in the following section.

8.        The description of the flow chart is too simple. More presentation is suggested.

9.        The meaning of the global variables should be discussed in detail.

10.    The font style of ℷ in line 304 seems not correct.

Reviewer 2 Report

In this work, the authors have presented a piezoelectric flag structure along with an SSHC interface circuit without inductor. 

Theory, simulation, and experimental results have been well presented overall. The concept of critical value of dt is also explained. Some comments are as follows:

1. Line 42: Should be 'pulse wiDth'

some other examples: Line 152 -- 'hard' instead of 'hardly'; Line 474: 'experiments' instead of 'experimental'.

Several other grammar and typo related corrections are needed throughout the manuscript.

2. Figs 32 and 33 can be clearer, so it is easier to read the values.

Reviewer 3 Report

In this paper, a piezoelectric flag was fabricated using PVDF and SEH (Standard Energy Harvesting) and SSHC (Synchronized Switch Harvesting on Capacitors) interface circuits were well explained. The theoretical and experimental contents of the design and operation of the proposed SSHC in this paper are well expressed. However, the following major and minor comments need to be supplemented and corrected, and if there is no sufficient explanation for this, it cannot be acceptable.

1.     It is necessary to explain what is the difference in the structure of the PVDF piezoelectric flag. The expression “A new structure of piezoelectric flag” was used in the abstract, but in the case of the main text, the structure of the piezoelectric flag that is not much different from a simple bimorph cantilever structure. More explanation is needed as to of what the merit of the structure. (Is the fabrication of a larger size flag different from other studies?)

2.     The simulation and experiment results of the piezoelectric flag were carried out at a fairly high wind speed. The wind speed above 18 m/s are not common situations, and it is difficult to apply the harvester in an actual environment.

3.     There are questions about the data in Figures 34-39. In the case of Figure 34 and Figure 35, the output performance according to SEH and SSHC was expressed through a three-dimensional graph, and the Z-axis data can be checked as voltage(V). However, in the case of Figures 36 to 39, the Y-axis values appears as power(W), and these values are confirmed to be consistent with Figure 34 and Figure 35. I hope this is a simple typographical error. In addition, in the case of a harvester, the output power value can be a major criterion for determining whether or not the output power can be used as an actual power source, and comparative evaluation with other studies is needed with the results.

4.     The spacing between numbers and units is incorrect. In general, a space between units and numbers is required, and although it may be a minor part, it should be checked as it can reduce the quality of the research.

Reviewer 4 Report

The authors report on a flag-type piezoelectric energy harvester from the wind. There are some issues to be taken care of throughout the manuscript. Besides, the inclusion of some more information would strengthen the quality of the paper. 

Here are my concerns:

11. The novelty of this work is unclear. In particular, the authors should clarify the novelty of this work.

22. [Abstract] should be more informative to present the findings of the work (including the results).

33.    [Introduction] Should be more organized and informative. The last paragraph of the introduction must be modified and updated with the novelty, scientific soundness, technical issues, and methods used, as well as quantified results of the proposed work. The current content does not have any adequate information.

44.   All figures' quality needs to be improved in this manuscript. Please merge the figure like Figure 1, 2, 3, 4 make one figure and indicate like (a), (b), (c)…. Figure 5 & 6; 7, 8, 9 &10, 11,12,13, &14; 16,17,18, &19; 20,21,22,23, &24…. The authors need to manage all figure in good shape.

55.  How about the optimized load resistance? In Figure 36, it's very difficult to see the optimized conditions. Why the sudden increase the power after 2000 kΩ in SEH? How do the authors calculate the power?

66. How do the authors optimize the wind speed? In the environment, where the proposed device will be kept for better performance?

Round 2

Reviewer 4 Report

The authors responded to all my comments very well and revised the manuscript accordingly. Now, I recommend the manuscript to be published in its current form.